# Phylogenetic Weighting Does Little to Improve the Accuracy of Evolutionary Coupling Analyses

**DOI:** 10.3390/e21101000

**Published:** 2019-10-12

**Authors:** Adam J. Hockenberry, Claus O. Wilke

**Affiliations:** Department of Integrative Biology, The University of Texas at Austin, Austin, TX 78712, USA; wilke@austin.utexas.edu

**Keywords:** direct coupling analysis, evolutionary coupling analysis, contact prediction, phylogenetic bias

## Abstract

Homologous sequence alignments contain important information about the constraints that shape protein family evolution. Correlated changes between different residues, for instance, can be highly predictive of physical contacts within three-dimensional structures. Detecting such co-evolutionary signals via direct coupling analysis is particularly challenging given the shared phylogenetic history and uneven sampling of different lineages from which protein sequences are derived. Current best practices for mitigating such effects include sequence-identity-based weighting of input sequences and post-hoc re-scaling of evolutionary coupling scores. However, numerous weighting schemes have been previously developed for other applications, and it is unknown whether any of these schemes may better account for phylogenetic artifacts in evolutionary coupling analyses. Here, we show across a dataset of 150 diverse protein families that the current best practices out-perform several alternative sequence- and tree-based weighting methods. Nevertheless, we find that sequence weighting in general provides only a minor benefit relative to post-hoc transformations that re-scale the derived evolutionary couplings. While our findings do not rule out the possibility that an as-yet-untested weighting method may show improved results, the similar predictive accuracies that we observe across conceptually distinct weighting methods suggests that there may be little room for further improvement on top of existing strategies.

## 1. Introduction

Correlated evolution of amino acid positions within a sequence alignment can be leveraged to inform structural models of proteins, predict mutational effects, and identify protein binding partners [1,2,3,4,5]. The ability to detect correlated evolution has been revolutionized by direct coupling analyses and other related methods that seek to re-construct one- and two-site marginal amino acid probabilities based on the observed distribution of sequence data [6,7,8,9,10,11]. Inference of two-site coupling parameters from a multiple sequence alignment is technically challenging, however, and numerous related approaches have been developed in recent years [9,10,12,13,14,15,16,17]. This intense focus on related methodologies stems from the fact that the highest scoring evolutionary coupling values are highly enriched in residue-residue pairs whose side-chains physically interact within three dimensional structures [18]. Evolutionary couplings can thus provide valuable information about structural constraints within and between protein families, while only requiring sequence information as inputs [15,19,20,21,22].

All methods to detect correlated evolution between different positions in a protein family require large numbers of representative sequences and therefore start by finding—and subsequently aligning—homologous sequences from large sequence databases [5]. An oft-remarked upon fact is that sequence databases are composed of a highly biased sample of life on Earth; some species are much more densely sampled than others (as are some genera, families, orders, etc.) [23,24,25,26,27]. Even if all extant life were equally well sampled and represented in sequence databases, species are related by complicated historical patterns and cannot be considered as independent observations [28].

Statistical issues arising from this shared phylogenetic history and biased sampling have long been noted by biologists [28]. The problem can be most clearly summarized by a toy example. In Figure 1A, we show a hypothetical sequence alignment and ask the question: What amino acid is preferred at the indicated site? At first glance, a phylogenetically agnostic method would simply count the frequency of different amino acids and conclude that valine (V, four occurrences) is preferred. However, accounting for phylogenetic relationships, a different perspective could reasonably conclude that threonine (T, three occurrences) is more highly preferred given that it occupies a substantially larger fraction of the phylogenetic tree and therefore dominates the evolutionary history of the protein family; the abundance of valines in the alignment is an apparent result of over-sampling one closely related lineage (which may represent numerous representatives of the same species, for example). Naively, the problem can be solved by simply selecting a single member from each species to prevent over-sampling. However, the issue remains equally problematic at other taxonomic levels (i.e., sampling numerous species from the same genus, numerous genera from the same family, etc.) and it is clear that a more general solution is required.

Prior research has shown that the best way to account for phylogenetic effects is to explicitly incorporate an evolutionary model into the statistical methods whenever possible [29,30,31,32,33,34,35,36]. However, this strategy can be challenging for certain problems [37] and simpler methods that differentially weight taxa according to their overall similarity to other taxa in a given dataset have been developed and applied for decades [38,39,40,41,42,43,44,45,46]. In the context of the toy example in Figure 1A, the choice of valine as the preferred amino acid comes from a model that weights each sequence uniformly. By down-weighting highly similar sequences, however, weighted frequencies could be used to come to the conclusion that threonine is instead the preferred amino acid. Instead of looking at preferred amino acid residues (one-site probabilities), evolutionary coupling analyses use sequence alignments to infer co-evolving positions via their two-site marginal probabilities. The current best practice for evolutionary coupling analyses is to down-weight sequences that are highly similar to one-another when inferring parameters from the multiple sequence alignment data. While this strategy appears in numerous methods, a systematic analysis of the benefit that sequence weighting provides in comparison to uniform weights, and an evaluation of different conceptually distinct strategies for assigning weights to sequences has not been performed to our knowledge.

Here, we evaluate existing weighting strategies alongside alternative tree- and sequence-based methods that have been proposed and used in various biological applications. We define the accuracy of a given method according to how well the resulting evolutionary couplings are able to predict residue–residue contacts within known representative structures of protein families [18]. Despite potential theoretical disadvantages, we find that the current best practice method of 80% sequence-identity-based weighting outperforms alternative methods that explicitly incorporate knowledge of phylogenetic relatedness. We show that a modification of this method provides a slight but insignificant improvement, and more broadly show that several methodologically distinct methods produce accuracies that are nearly indistinguishable both from one-another and from uniform weights.

## 2. Results

### 2.1. An Explanation of Weighting Methods

Many variants of evolutionary coupling analysis methods have been developed, and most methods implement a sequence-identity-based correction to mitigate the effect of phylogenetic relatedness [10,11,13]. Specifically, given *n* sequences in an alignment, the pairwise similarity of all sequences is calculated and the weight W(i) of a given sequence *i* within an alignment equals the inverse of the total number of sequences *j* whose distance d(i,j) to sequence *i* is less than some parameter λ:(1)$$W\left(i\right)=1/\sum_{j=1}^{n}I\left(i,j\right),$$
where *n* is the number of sequences in the alignment and I(i,j) is an indicator variable defined as
(2)$$I\left(i,j\right)=\left\{\begin{array}{ccc} 0 & if & d_{i,j}<\lambda ,\\ 1 & if & d_{i,j}>=\lambda . \end{array}\right.$$

The distance d(i,j) and the cutoff λ are usually measured as percent sequence identity: the number of identical residues between two aligned sequences divided by their total length.

Under this weighting scheme, highly unique sequences are given a weight value of 1, whereas sequences that are similar to others are assigned weights between 0 and 1 according to how many such similar sequences are in the alignment. Given this strategy, the effective number of sequences is simply the sum the weights assigned to all sequences, which takes a value between 0 and *n*.

Several possible issues arise from this weighting scheme. First, it is not immediately apparent what value of λ is most appropriate to use as a sequence identity threshold. While this parameter can be optimized for practical utility (the field has coalesced largely around a value of 80%), it is unclear what this value tells us about the co-evolutionary process or *why* it works so well. Second, this weighting scheme can produce some counter-intuitive results. Given an 80% sequence identity threshold, two otherwise independent sequences in an alignment sharing 99% sequence identity will each be assigned a weight of 0.5 reflecting their relative similarity to one another. In the same alignment, two sequences sharing 81% sequence identity will similarly each be assigned a weight of 0.5 despite being much more distinct from one another compared to the former pair. However, two sequences sharing 79% sequence identity will be assigned a weight of 1.0. Finally, the underlying phylogenetic history of the sequence evolution is ignored by this sequence-based comparison method which may inhibit its overall effectiveness.

Our goal here is not to exhaustively evaluate all possible strategies for assigning weights to sequences or tips on a phylogeny but rather to test several popular methods that represent logical starting points for possible improvements for use in evolutionary coupling analyses. Specifically, we decided to implement and test three algorithms: one sequence-based method and two conceptually distinct tree-based methods. The sequence-based method was proposed by Henikoff and Henikoff [44] and proceeds across each position by first awarding each observed residue at given position in an alignment an equal share of the weight for that position (where each position in the alignment has a starting weight of 1). The weights at that position for each sequence in the alignment are then assigned by dividing the weight assigned to each residue equally among all sequences sharing the same residue. Finally, the weight of a given sequence is simply the sum of the weights assigned to each position/residue. The method gives intuitively correct results for toy examples and has been used in numerous popular applications including HMMER and PSI-BLAST, with several different modifications for dealing with gap sequences [47,48].

We additionally implemented two tree-based methods that were initially proposed by Altschul et al. [38] (hereafter referred to as “ACL” weights) and Gerstein et al. [43] (hereafter referred to as “GSC” weights). The ACL method is equivalent to a model of electricity where a power source is plugged into the root of the tree, each branch provides resistance proportional to its length, and the current flowing out of each tip is used to determine the weights [38]. By contrast, the GSC method is a way of partitioning the branch lengths of a tree where the final weight of each tip is a weighted sum of all the branch lengths leading up to it [38,43]. Conceptually, ACL and GSC weights are quite distinct with GSC weights assigning a higher weight to tips that have particularly long branch lengths (and thus occupy a larger proportion of the tree) and ACL weights assigning the highest weights to sequences with particularly short branch lengths that reside closest to the root. We note that both metrics explicitly account for the underlying tree topology and thus require a previously constructed rooted evolutionary tree.

A notable caveat to the HH, ACL, and GSC weighting methods is that they do not provide intuitive *absolute* scales. The sum of all HH weights in their original formulation is equivalent to the length of the alignment, ACL weights are relative and sum to 1, and GSC weights are in units of branch length (substitutions per unit time) [38,43,44]. Differences in absolute scales will affect the output of co-evolutionary models because the regularization strength used during model fitting is proportional to the number of effective sequences. That is to say, a model fit to data where all weights are assigned a uniform value of 1 will be different from a model fit to the same set of sequences where all weights are assigned a uniform value of 0.1. Thus, both the relative differences in weights and their absolute scale are important considerations. For each of the three new methods, we employ two re-scaling strategies: First, we divide each weight value by the mean for that alignment, such that the weights for a given alignment will sum to *n*, where *n* is the number of sequences. Second, we divide each weight by the maximum observed weight in an alignment, such that the largest relative weight will be assigned a value of 1 and all other weights are some fraction of this.

For an example protein (PDB:1AOE), assigning weights to a sequence alignment/tree demonstrates that the methods vary substantially in how uniformly they distribute weights (Figure 1B). The GINI coefficient is a measurement of uniformity where values of zero correspond to uniform weights and values approaching 1 illustrate the case where a small number of sequences have very large weights while the remainder have very small weights. This relationship can be visualized by a Lorenz curve, which in this case plots the cumulative fraction of weights (y-axis) against the cumulative fraction of sequences (x-axis, sorted from lowest to highest weights). The Lorenz curves in Figure 1B show that ACL weights in particular result in a highly uneven distribution of weights. This finding holds more broadly across a dataset of 150 diverse protein families; the tree-based methods produce a more un-even distribution of weights, with ACL weights being particularly highly skewed (Figure 1C).

The different weighting schemes (when applied to the same multiple sequence alignment) are only modestly correlated with one-another. Figure 1D shows the median correlation (across the 150 protein families) observed among HH, GSC, and ACL as well as the commonly used 80% sequence-identity-based re-weighting method. In general, the weights produced by different methods on the same protein family are significantly positively correlated with one-another, but the correlations are fairly low, demonstrating that the weighting methods themselves are distinct. We additionally note that tree-based weighting methods may be subject to numerous errors during the tree construction and rooting process. We performed bootstrap resampling of multiple sequence alignments to compare the resulting weights to the originally calculated set of weights and found that they were significantly positively correlated (Appendix A). GSC weights, however, were much more robust (median Spearman’s ρ of 0.84) compared to ACL weights (median Spearman’s ρ of 0.61).

### 2.2. Sequence Weighting Does Little to Improve Contact Predictions

To test the effectiveness of different weighting methods, we calculated evolutionary couplings using the program CCMPredPy—a Python-based implementation of one of the most popular pseudo-likelihood based methods (CCMPred), which we modified to accept weights from externally supplied files—for 150 unique protein families with known structural representatives [13,16]. We next tested what fraction of the top *L* couplings for a given protein family (where *L* is the length of the reference sequence with a known three-dimensional structure) are true intramolecular residue–residue contacts—a metric known as the Positive Predictive Value (PPV) (see Section 4 for details) [18]. We separately quantified accuracies from the raw evolutionary couplings, entropy-corrected couplings, and Average Product Corrected (APC) couplings. The latter two post-hoc corrections have been shown to improve the accuracy of evolutionary couplings by accounting for uneven sequence entropies across positions in the alignment and perhaps the underlying phylogenetic structure [16,49].

As expected, we found that across all weighting schemes, the APC (and to a slightly lesser extent, the entropy-corrected) evolutionary couplings produce substantially more accurate results compared to raw coupling scores (Figure 2). In nearly all cases, sequence-identity-based weighting resulted in the highest accuracy. For the best performing APC coupling scores (Figure 2A), the commonly used λ parameter representing an 80% sequence identity threshold resulted in significantly higher accuracies compared to the uniform weight controls (Wilcoxon signed-rank test, p<0.001). One phylogeny-based weighting method (GSC) and the HH sequence-based method were slightly more accurate than uniform weights provided that they were mean-scaled but the improvement was not significant in either case (p=0.09 and p=0.1, respectively); both methods were significantly less accurate than the 80% sequence-identity-based method (p<0.001 for both cases). ACL weights by contrast generally performed poorly in all cases.

We note that even in the best case scenario the increase in PPV due to sequence weighting is comparatively small when compared to the large improvements in accuracy that result from the post-hoc APC and entropy corrections: median PPV for uniform weights are more than twice as high for APC couplings relative to raw couplings. Interestingly, the best performing weighting schemes substantially improve the accuracy of raw evolutionary couplings relative to the uniform weight control (Figure 2C, 44% median increase in PPV for max-scaled GSC weights, p<0.001), but do comparatively little in the case of the more accurate APC couplings (Figure 2A, 2% median increase in PPV for 80% sequence-identity-based weights, p<0.001).

A caveat noted above is that the regularization strength of the CCMPred model is proportional to the effective number of sequences. The typical value used for the pairwise regularization strength parameter (“LFACTOR”) is 0.2, but this regularization strength was tuned for the previously best performing 80% sequence identity-based weights that are commonly employed. For the GSC, ACL, and HH methods, we tested a range of parameters (from 0.05 to 1.0) to see if a different pairwise regularization strength parameter might produce superior results (Appendix A). No combination of weighting method and parameter values, however, results in substantially improved accuracy for the best performing APC couplings. For all of the max scaled methods, smaller values of this parameter substantially improve results but only up to the level achieved by the best performing mean-scaled methods. Perhaps most notably, for entropy-corrected couplings, we found that larger pairwise regularization strength parameters were helpful for the best performing methods and brought the overall mean PPVs nearly on par with that of the APC couplings (Appendix A). The strength of regularization is thus an important consideration when evaluating different weighting schemes, but our finding that numerous methods achieve roughly the same overall accuracy remains unchanged from this analysis.

### 2.3. Weighting on Time-Scaled Trees

In Figure 1, we note that tree-based weighting methods produced a more un-even distribution of weights compared to the sequence-based weighting methods that we tested. A potential issue with both of the tree-based weighting methods that we consider here (aside from the possible noise/error in their calculation that was previously noted) is that the rates of evolution vary across phylogenetic trees and thus species are not equidistant from the root sequence. Phylogenetic trees reflect both the relationship between species and the rate of evolution along each branch. For trees consisting solely of extant species, numerous methods can re-scale trees to produce tips that are contemporaneous and equidistant from the root (Figure 3A) [50]. Since GSC and ACL weighting methods are significantly influenced by the overall distance from the root for individual tips, we reasoned that computing these weights on scaled-trees may produce less variable weights and perhaps more accurate results. We thus used the RelTime algorithm to transform each raw tree into a time-scaled tree and re-computed the weights for the two tree-based weighting methods on these RelTime trees [50].

For a given protein alignment, weights constructed in this manner display significantly less heterogeneity than weights calculated from the raw trees (Wilcoxon signed-rank test, p<0.001). The PPVs of mean- and max-scaled weighting methods were significantly improved in all cases relative to weights computed on the raw trees (Figure 3B, results shown for APC couplings). The improvements were again comparatively small and no method out-performed 80% sequence-identity-based weights. However, PPVs with mean-scaled GSC weights calculated from RelTime trees were significantly higher than PPvs from uniform weighting (Wilcoxon signed-rank test, p=0.003) and the difference in PPV between these weights and the best performing 80% sequence-identity-based weights was not significant (p=0.14).

### 2.4. An Altered Sequence-Identity-Based Method That Accounts for Sequence Similarity

Thus far, we have shown that the current best practice of using sequence-identity-based weighting within a 80% sequence similarity neighborhood results in evolutionary couplings that have the highest power to predict intra-molecular residue–residue contacts. However, we also discussed some potentially counter-intuitive properties of this sequence-identity-based method. We thus developed and tested a variant of the sequence-identity-based method that down-weights sequences according to pairwise similarity and an identity threshold, but does so by accounting for the actual similarity between the sequences. Whereas the original method assigns each sequence a value of 1 and divides by the raw number of similar sequences (defined according to the λ parameter), our modification instead divides by the sum of a similarity-adjusted value for each sequence. Specifically,

(3)$$W\left(i\right)=1/\sum_{j=1}^{n}I_{\mathrm{adj}}\left(i,j\right).$$

In contrast to Equation (Equation 2), $I_{\mathrm{adj}}\left(i,j\right)$ produces a continuous range of values between 0 and 1: (4)$$I_{\mathrm{adj}}\left(i,j\right)=\left\{\begin{array}{ccc} 0 & if & d_{i,j}<\lambda ,\\ \left(d_{i,j}-\lambda \right)/\left(1-\lambda \right) & if & d_{i,j}>=\lambda . \end{array}\right.$$

As in Equations (Equation 1) and (Equation 2), the distance $d_{i,j}$ and the cutoff λ are measured as percent sequence identity.

Using this method with a λ value of 0.8, two otherwise independent sequences in an alignment with 99% sequence identity will each be assigned a weight of 0.513 [or 1/(1+0.95), where 0.95=(0.99-0.8)/(1-0.8)], reflecting their high similarity to one another. In the same alignment, two sequences sharing only 81% sequence identity will by contrast each be assigned only a slightly decreased weight of 0.95 [or 1/(1+0.05), where 0.05=(0.81-0.8)/(1-0.8)]. All else being equal, the more similar sequences are, the more they will be down-weighted up to the given sequence identity threshold, at which point no further down-weighting occurs.

Comparing this similarity-adjusted sequence-identity-based method to the original method shows that the similarity-based adjustment produces more robust results across the range of possible values for λ (Figure 4). Across all of the different variants that we tested, similarity-adjusted sequence-identity-based weights with an identity parameter of 0.8 (and the APC, Figure 4A) produced evolutionary couplings with the highest median and mean PPV for the 150 protein families. PPVs resulting from this method were significantly higher than results from uniform weights (1.9% median and 3.7% mean increase in PPV, Wilcoxon signed-rank test p<0.001) but the increase compared to 80% sequence-identity weights calculated in the original manner was slight and not significant (0% median and 0.3% mean increase in PPV, p=0.11).

## 3. Discussion

Natural sequence alignments are not composed of independently evolved lineages and instead have an unknown pattern of relationships that can be inferred and visualized as a phylogenetic tree. Statistical methods that fail to account for these relationships are expected to be biased, but in the case of direct coupling analyses a phylogenetically agnostic model has nevertheless proven valuable at predicting residue–residue contacts within protein structures [5,10,11]. Differential sequence weighting is commonly employed in such analyses as a way to partially mitigate phylogenetic effects, but the overall benefit that such weights provide has yet to be systematically interrogated. We have shown here that numerous (and conceptually distinct) weighting methods produce evolutionary couplings with a roughly equivalent ability to predict residue–residue contacts—given that the coupling values are transformed post-hoc via the average product correction (APC). We found that uniform, HH, GSC, and two variants of 80% sequence-identity-based weights all produce nearly indistinguishable accuracies from one another. While we have only evaluated a few different weighting methods and variants, the similar predictive power of top-performing weighting strategies (despite being substantially different from one-another, Figure 1D suggests that there may be little room for improvement on top of current best practices.

Intuitively, uneven sampling and phylogenetic biases are *expected* to introduce spurious effects into statistical models. Indeed, this is known to be the case in numerous contexts, such as when assessing the strength of correlations between discrete and continuous traits [28,34,36]. Nevertheless, we have shown here that using variable sequence weights to correct for these problems provides little (if any) practical benefit when attempting to predict residue–residue contacts. Why might this be the case? We caution that weights alone are an imperfect method of accounting for shared phylogenetic history, and in other contexts achieving accurate true and false positive rates from statistical tests requires more than simple re-weighting of data points [29,31,36,51,52].

In the context of evolutionary couplings, it is unclear whether uneven sampling and phylogenetic biases do not affect the fitting of coupling parameters as much as one might initially think, whether the APC (a post-hoc re-scaling procedure) largely corrects for any such factors, or whether weighting in general is simply an inadequate solution to the problem of phylogeny. Several lines of evidence currently indicate that the overall effect of phylogeny in direct coupling analysis models may be minimal. For instance, our results confirm previous findings showing that correcting for column-wise entropy produces comparable accuracies when compared to the average product correction (even though the latter is thought to partially correct for phylogenetic effects) [16]. Recent work has also shown that eigenvectors with the largest eigenvalues in a residue–residue covariance matrix strongly reflect the phylogenetic relatedness of the aligned sequences [53,54,55]. Removal of these eigenvectors substantially improves predictions of structural contacts in this conceptually distinct model, and variants of direct coupling analysis appear to achieve this same result via different means [55]. Future studies investigating the contribution of uneven sequence weighting towards these high value eigenvectors may be particularly illustrative about the impact of phylogenetic weights and their potential role in covariation analyses moving forward.

While we found that numerous weighting methods produce roughly equivalent end results on average, our findings raise other several potential issues that may be worthy of further study moving forward. We noted that many weighting methods do not clearly provide an intuitive absolute scale and instead assign weights to sequences (or tips in a phylogenetic tree) that are either relative or in irrelevant units. This can be problematic from a practical standpoint because most methods for inferring evolutionary coupling parameters between residue–residue pairs rely on some form of prior and the weight given to observed data relative to this prior may affect results. For the HH, GSC, and ACL methods, we found that two different scaling procedures (which maintain relative weights within a dataset but change their absolute values) produced varying accuracies (Figure 2). With the exception of star phylogenies, the effective sample size from phylogenetically structured data is strictly less than the number of sequences/data points analyzed. More accurately estimating the effective sample size and scaling weights accordingly may improve the performance of different weighting schemes beyond what we observed here.

Additionally, the HH, GSC, and ACL methods do not include a free parameter that can be tuned to improve results. We validated that an 80% sequence-identity neighborhood is optimal using the currently accepted method and a similarity-adjusted variant, but this 80% value is a free parameter that has been optimized to produce the highest accuracy for sequence-identity-based weighting. What we believe the optimality of this parameter represents in practice is that once two sequences diverge past approximately 80% similarity, their evolution is effectively independent. If this is the case, down-weighting sequences that for instance share 50% sequence identity might make little sense. Fully answering this question, however, would require testing a range of regularization strengths since the effective number of sequences at that level of sequence identity down-weighting is substantially less than at 80% sequence identity. By contrast, the HH, GSC, and ACL methods all inherently compare each sequence to every other sequence in a global manner. It seems possible that some phylogenetic tree transformation may be able to introduce the same intuition of ignoring evolutionary relatedness past some threshold level into tree-based weighting methods [30,32]. The best way to perform such re-scaling, or how to perform something conceptually similar for HH weights, is a promising area for future research.

Another potential area for future research is to specifically investigate the cases when sequence re-weighting makes the largest/smallest impact on PPV relative to uniform weights. The overall predictive power of evolutionary couplings varies both within and between protein families. Previous work has shown that within an individual protein family, structural contacts that are mediated by side-chain interactions are most likely to be detected by co-evolutionary methods, as opposed to those mediated by atomic interactions in the peptide backbone [18]. Between protein families, the substantial variability in PPVs result from numerous factors including but not limited to the number of sequences in an alignment, the diversity of those sequences, homo-oligomerization, alignment errors from repeat proteins, and family structural variation [56]. While we found that PPVs are highly variable across protein families and that sequence re-weighting can help increase these scores a relatively small amount on average (Figure 2), the magnitude of this increase is higher for some families compared to others. Being able to associate the magnitude of the increase with properties of the sequences or tree, such as their diversity or bias, may provide interesting clues about the general ineffectiveness of sequence weighting or insight into novel strategies that could better account for these effects.

Despite being weakly correlated with one another, uniform, 80% sequence identity, HH, and GSC weights perform roughly equivalently at predicting residue–residue contacts. We recommend that any method with substantially improved performance should become the standard but computational complexity and run-time are real constraints that most researchers should additionally consider. Once a phylogenetic tree is constructed, the cost of calculating the different weights that we considered here is negligible relative to the run-time of inference algorithms. However, given the ideal size of protein family alignments (thousands to tens of thousands of sequences), the most accurate methods for phylogenetic tree construction are computationally infeasible. Even more rapid methods, such as those we employed here, may substantially increase the overall run-time for a pipeline relying on tree-based weights. At present, our current results give no indication that the increased computational time and complexity of tree-construction will provide any benefit. If we were to see such a benefit, the choice of whether a few percent increase in accuracy would be worth doubling (or worse) the run-time for a protein family of interest would be dependent on the researcher and the application.

While several methods were nearly indistinguishable from one another in terms of their accuracies, we did find that a slightly modified sequence-identity-based re-weighting method that accounts for sequence similarity actually performs the best of any method that we tested. This method does not require calculation of a phylogenetic tree and therefore has an overall run-time that is virtually equivalent assuming uniform weights or existing best practices. However, using either the original or similarity-adjusted sequence-identity-based weighting can be expected to offer less than a few percent improvement in accuracy compared to uniform weights, which completely ignore phylogeny. We therefore speculate that if phylogenetic effects are truly problematic for inferring co-evolution—and we caution that this is not necessarily a given—then substantial improvements to existing methods may require the explicit incorporation of phylogenies and time-dependent sequence evolution rather than heuristic re-weighting strategies.

## 4. Materials and Methods

### 4.1. Description of the Dataset

For all of our analyses, we used the so-called “psicov” dataset—an existing set of 150 distinct protein structures with corresponding multiple sequence alignments that have been used in numerous benchmark studies for predicting residue–residue contacts from evolutionary couplings [14,57,58]. All sequence and structure data were taken directly from Jones and Kandathil [58], but, given the large number of different analyses that we ran, we first randomly down-sampled each alignment to a maximum of 1001 sequences (1000 sequences plus the mandated inclusion of the reference protein sequence).

### 4.2. Phylogenetic Tree Construction

For each sequence alignment in our dataset, we constructed a rough phylogenetic tree using the double precision version of FastTree2 (v.2.1.10; LG model, gamma distributed rate variation, with pseudocounts) [59]. We next adjusted the branch lengths on each guide tree by running the alignment and the template tree through the more accurate IQtree software (v1.6.9; LG model, Gamma-distributed rate variation with 20 categories) [60]. Finally, we rooted the resulting trees using the mid-point method [61].

For RelTime trees, we implemented our own version of the RelTime algorithm as described in the original manuscript while ensuring that our method produced similar results [50]. We note here only that our implementation does not perform a statistical test (and subsequent alteration of rates) at the end of the algorithm to ensure that rate changes are significant.

### 4.3. Weighting Methods

We developed all of our weighting methods from scratch using custom python programs that heavily leveraged tools from the Biopython package [61]. For sequence identity weighting and the novel similarity-adjusted version we propose here, details are presented in the main text, Equations (Equation 1)–(Equation 4). We ensured that our own version of sequence-identity-based weighting was equivalent to the method implemented within CCMpredPy by comparing the resulting effective number of sequences metrics and accuracies and finding them to be identical.

For HH based weights, we followed the procedure outlined in the initial paper and ensured that our implementation gave the desired results on the toy examples presented therein [44]. Researchers have pointed out subsequent modifications to this method [47,48] concerning how to effectively treat gap sequences. Rather than treating these as a 21st character as some implementations have done, our implementation assigns gap sequences a weight value of zero. Further, each column in the alignment is weighted from 0 to 1 according to the fraction of non-gapped positions. In this manner, alignment positions with more gaps are assigned lower weights and the positions with gaps themselves contribute a weight of zero. Summation and calculation of final weights follows the published procedure [44]. However, since the units and absolute value of these weights are not intuitive, we finally re-scaled the weights via separate mean- and max-scaling procedures. In mean-scaling, we calculate the mean of all weights determined via the HH algorithm for a particular sequence alignment and then divide the weight of each sequence in the alignment by this value. This ensures that the sum of all final weights will be equal to the number of sequences in the alignment (*n*). In the separate max-scaling procedure, we find the maximum weight observed for a particular sequence alignment, and subsequently divide all weights in the alignment by this value. The sum of all weights following this procedure is guaranteed to be some value less than the total number of sequences (*n*).

For ACL and GSC weights, we again followed the procedures outlined in the respective manuscripts [38,43] and ensured that our implementations produced identical results to the examples presented therein. As with HH, calculation of final weights occurred by (separately) scaling the weight values via their mean and maximum values as noted above.

### 4.4. Evolutionary Coupling Analysis

We chose to use CCMpredPy (v1.0.0, contained as part of the CCMgen package) [13,16] for all evolutionary coupling analyses since we were able to modify the source code for this popular method to accept externally supplied weights in the form of a simple text file where the weight value for each sequence corresponded to its line in the input sequence file. We used the default values with the ofn-pll flag corresponding to the pseudo-likelihood optimization of coupling parameters. For each different weighting method that we tested, we outputted files corresponding to the raw, entropy-corrected, and average product corrected coupling matrices.

### 4.5. Structural Analysis and Accuracy Determination

We used the .PDB files provided as part of the psicov dataset and for each structure computed a matrix of residue–residue distances. Each distance value is measured according to the geometric center for all side-chain heavy atoms for a particular residue (including the Cβ atom, excluding the Cα atom) [18]. In the case of glutamine, the side-chain center coordinates were assigned to the Cα atom. We determined residue–residue contacts according to a uniform 7.5 angstrom threshold for all proteins.

We determined the accuracy of evolutionary couplings by determining how well they were able to predict residue–residue contacts within a reference structure. We first selected the top *L*-ranked couplings for each dataset, where *L* corresponds to the length of the reference protein sequence (i.e., the sequence for which we have a known structure). The PPV for a particular dataset corresponds to the fraction of those top *L*-ranked couplings that are classified as residue–residue contacts according to the above definition.

## Figures and Tables

**Figure 1 entropy-21-01000-f001:**
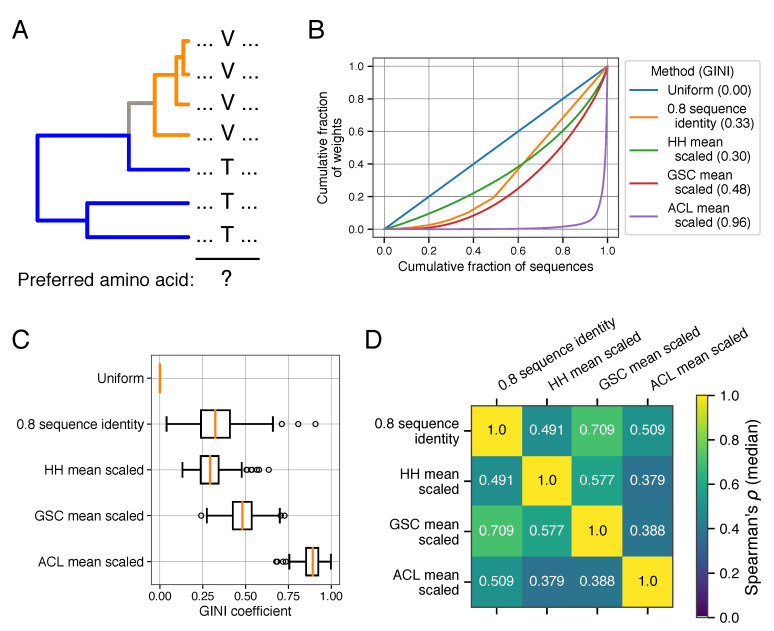
Weighting methods and their relationships in empirical datasets. (**A**) A toy example illustrating the problem of biased sampling and phylogenetic relatedness. Judging by their frequency (i.e., uniform weighting), valine (V) is the preferred amino acid at the indicated position. However, threonine (T) occupies a substantially larger proportion of the inferred evolutionary history. (**B**) For an example protein sequence alignment (PDB:1AOE), different weighting strategies produce a more- and less-uniform distribution of weights as visualized by the Lorenz curve. (**C**) The distribution of GINI coefficients for 150 protein families (higher coefficients correspond to a less uniform distribution of weights) using different weighting strategies (boxes span the 25th to 75th percentiles, and the red line indicates the median). (**D**) The median correlation coefficient (Spearman’s ρ) of different weighting methods observed across the same 150 protein families.

**Figure 2 entropy-21-01000-f002:**
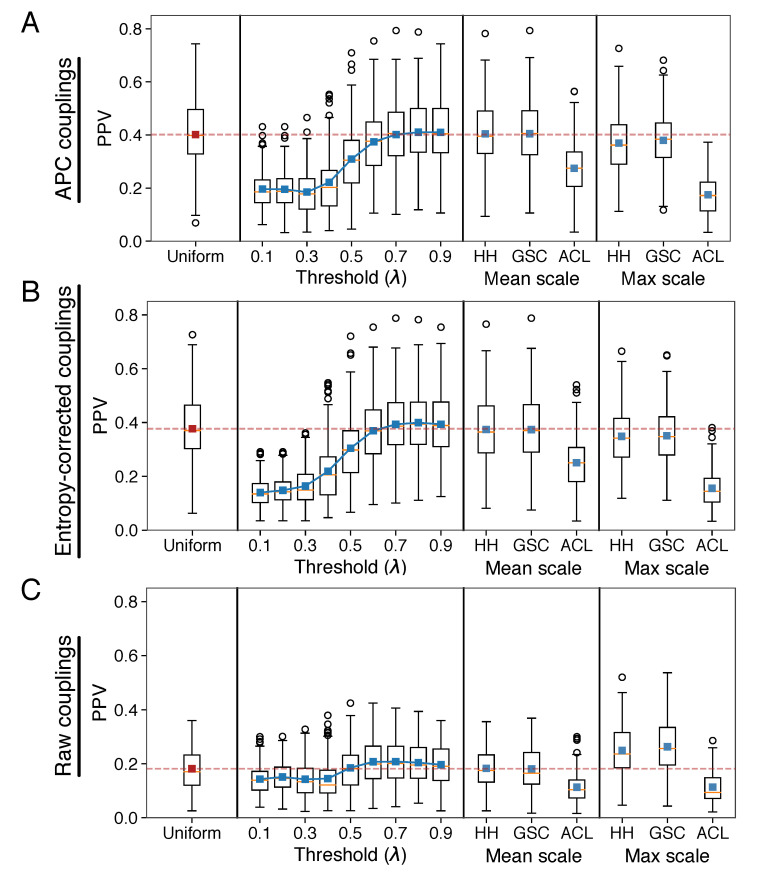
Testing the ability of evolutionary couplings to predict residue–residue contacts in representative structures. “Uniform” refers to the use of uniform weights for all sequences when fitting evolutionary coupling parameters (the red dashed line indicates the mean of this distribution and represents a baseline performance that methods should improve upon). “Threshold (λ)” refers to sequence-identity based weighting with different parameters, and “Mean scale” and “Max scale” refer to two different scalings of the indicated weighting methods (HH, GSC, and ACL). (**A**) Using APC couplings, the mean positive predictive values (PPVs) of the top *L* couplings vary across different weighting schemes used to infer evolutionary couplings. However, the only methods that significantly improve performance is sequence-identity-based re-weighting with λ = 0.8 or 0.9 (Wilcoxon signed-rank test, p<0.001), but the magnitude of the improvement is modest (1.9% and 1.1% median improvement over uniform). (**B**) Using entropy-corrected evolutionary coupling values leads to similar conclusions that no weighting scheme substantially outperforms uniform weights. (**C**) Using raw evolutionary coupling values results in substantially higher accuracies for certain weighting methods relative to uniform, but the overall accuracies remain low compared to (**A**,**B**).

**Figure 3 entropy-21-01000-f003:**
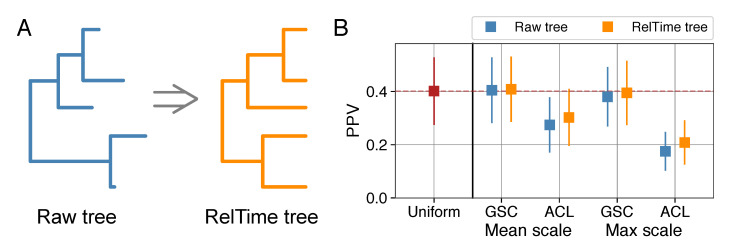
Tree re-scaling prior to calculation of weights slightly improves accuracies. (**A**) Raw, rooted phylogenetic trees can be converted to time-scaled trees with contemporaneous tips using the RelTime algorithm. (**B**) Sequence weights calculated from RelTime trees result in slightly better residue–residue contact prediction for the two tree-based weighting methods that we consider (and the two separate scalings of those weights). Shown is the mean PPV for 150 protein families using APC couplings, with error bars showing the standard deviation.

**Figure 4 entropy-21-01000-f004:**
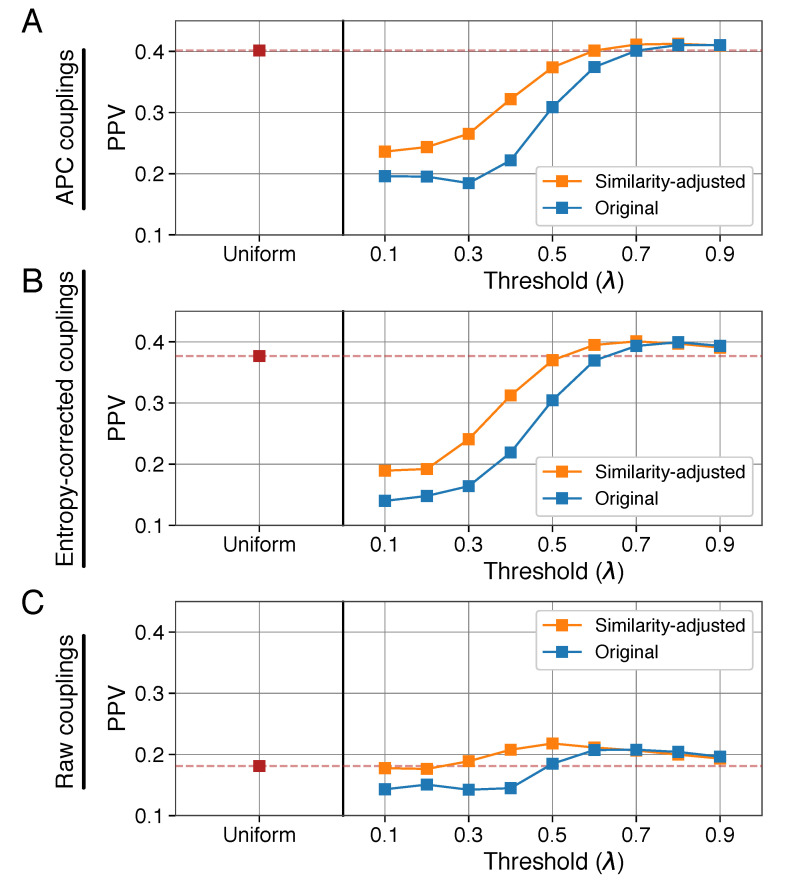
An altered sequence-identity-based method is more robust to parameter choice. (**A**) Using APC couplings, mean PPVs for similarity-adjusted sequence-identity-based weights are equal to or higher than PPVs calculated with the commonly used sequence-identity-based weights. (**B**) Same as in (**A**), using entropy-corrected evolutionary coupling values. (**C**) Same as in (**A**,**B**), using raw evolutionary coupling values.

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
