# Peer review of "Phylogenetic Weighting Does Little to Improve the Accuracy of Evolutionary Coupling Analyses"

_entropy, 2019, doi:10.3390/e21101000_

Round 1

Reviewer 1 Report

I found this study clear and well-written. I recommend for publication with minor revisions. I have heard anecdotal, unpublished reports to this end, and it is nice to see a careful study on its way for publication.

Comments:

1) Research has given some hints as to why correcting for phylogeny has not mattered as much as one would expect. The first paper is: "From principal component to directed coupling analysis of coevolution in proteins: low-eigenvalue modes are needed for structure prediction", the second is "Population structure and eigenanalysis", and a recently published paper that puts these two points together is "Power law tails in phylogenetic systems". The authors are Cocco, Monasson and Weigt; Patterson, Alkes, and Reich, and Qin and Colwell, respectively.

2) I'm curious how the phylogeny correction method proposed by this last paper by Qin and Colwell stacks up to APC. It would be interesting if this method still does not do better. I suggest that the authors add this comparison to their paper.

3) Why not report the computational complexity of these various corrections? I suppose they should all be dwarfed by the complexity of the inference algorithm, but tradeoff between scalability and accuracy should be addressed. In this 'big data' era, I think approximate methods that can eat a lot of data often outperform more exact methods that don't scale as well.

4) For this reason, I'm skeptical of the last sentence of the paper: "We therefore speculate that substantial improvements to evolutionary coupling analyses will require the explicit incorporation of phylogenies and time-dependent sequence evolution, but how to do so remains elusive." If explicitly incorporating phylogenies and sequence evolution is computationally expensive, it may not be worth the price. To give an example, Rivas and Eddy have recently published a study, "Parameterizing sequence alignment with an explicity evolutionary model" in which adding an evolutionary model had surprisingly little benefit while being slower. These points are worth discussing.

Cheers

Author Response

Response to reviewer 1:

Comments to the Author

I found this study clear and well-written. I recommend for publication with minor revisions. I have heard anecdotal, unpublished reports to this end, and it is nice to see a careful study on its way for publication.

We thank the reviewer for their kind words and for their constructive comments. Below we address each issue in turn.  

1) Research has given some hints as to why correcting for phylogeny has not mattered as much as one would expect. The first paper is: "From principal component to directed coupling analysis of coevolution in proteins: low-eigenvalue modes are needed for structure prediction", the second is "Population structure and eigenanalysis", and a recently published paper that puts these two points together is "Power law tails in phylogenetic systems". The authors are Cocco, Monasson and Weigt; Patterson, Alkes, and Reich, and Qin and Colwell, respectively.

We agree that all three of these references are very relevant to our work and thank the reviewer for noting their previous omission. We have added a paragraph in the discussion that explicitly highlights and references this work, and better contextualizes our findings:

“Recent work has also shown that eigenvectors with the largest eigenvalues in a residue–-residue covariance matrix strongly reflect the phylogenetic relatedness of the aligned sequences  [53–55]. Removal of these eigenvectors substantially improves predictions of structural contacts in this conceptually distinct model, and direct coupling analysis appears to achieve this same result via different means [55]. Future studies investigating the contribution of uneven sequence weighting towards these high-value eigenvectors may be particularly illustrative about the impact of phylogenetic weights and their potential role in covariation analyses moving forward. ”

2) I'm curious how the phylogeny correction method proposed by this last paper by Qin and Colwell stacks up to APC. It would be interesting if this method still does not do better. I suggest that the authors add this comparison to their paper.

While we agree that this paper is interesting and clearly relevant, we have opted not to include this analysis at this stage. We feel that comprehensively evaluating the role of uneven sequence weights in this conceptually distinct model could (and perhaps should) be the subject of a paper unto itself, that the results would warrant substantial space and therefore may detract from the succinct message of this manuscript. In the paragraph noted in response to comment #1, however, we have explicitly stated that this could/should be an area for future research and fully agree with the reviewer about its potential importance.

3) Why not report the computational complexity of these various corrections? I suppose they should all be dwarfed by the complexity of the inference algorithm, but tradeoff between scalability and accuracy should be addressed. In this 'big data' era, I think approximate methods that can eat a lot of data often outperform more exact methods that don't scale as well.

This is a very valid point and we have added a brief discussion of the issue of time and computational complexity to the discussion. Overall, the reviewer is correct that the inference algorithm dominates the run-time, but making phylogenetic trees from potentially tens of thousands of sequences is by no means trivial. In fact, we used a rapid heuristic tree construction algorithm here for exactly this reason (FastTree) because more accurate fully maximum likelihood (or Bayesian) tree construction algorithms (IQtree, RAxML, BEAST, etc.) are simply untenable at this scale. We note these issues in the following paragraph:  

“We recommend that any method with substantially improved performance should become the standard but computational complexity and run-time are real constraints that most researchers should additionally consider. Once a phylogenetic tree is constructed, the cost of calculating the different weights that we considered here is negligible relative to the run-time of inference algorithms. However, given the ideal size of protein family alignments (thousands to tens of thousands of sequences), the most accurate methods for phylogenetic tree construction are computationally infeasible. Even more rapid methods, such as we employed here, may substantially increase the overall run-time for a pipeline relying on tree-based weights. At present, our current results give no indication that the increased computational time and complexity of tree-construction will provide any benefit. If we were to see such a benefit, the choice of whether a few percent increase in accuracy would be worth doubling (or worse) the run-time for a protein family of interest would be dependent on the researcher and the application.”

4) For this reason, I'm skeptical of the last sentence of the paper: "We therefore speculate that substantial improvements to evolutionary coupling analyses will require the explicit incorporation of phylogenies and time-dependent sequence evolution, but how to do so remains elusive." If explicitly incorporating phylogenies and sequence evolution is computationally expensive, it may not be worth the price. To give an example, Rivas and Eddy have recently published a study, "Parameterizing sequence alignment with an explicity evolutionary model" in which adding an evolutionary model had surprisingly little benefit while being slower. These points are worth discussing.

This is an important point and we largely agree with the reviewer sentiments here and we have re-worded this paragraph accordingly. Namely, if phylogeny is important then we feel that explicitly incorporating the concept of time-dependency will be essential (as opposed to approaches like variable sequence weighting). But based on previous findings, we agree that it remains to be seen whether phylogeny is really much of a problem at all. We agree that run-time and computational complexity is an important consideration to be made on a case-by-case basis, and direct the reviewer to the paragraph that we added in response to comment #3 where we address the possible tradeoff in accuracy and run-time. We additionally re-worded the last line of our manuscript to reflect this reviewer concern:

“We therefore speculate that if phylogenetic effects are truly problematic for inferring co-evolution—and we caution that this is not necessarily a given—then substantial improvements to existing methods may require the explicit incorporation of phylogenies and time-dependent sequence evolution rather than heuristic re-weighting strategies.”

Reviewer 2 Report

It is a very useful study to benchmark the performance of various methods for detecting evolutionary couplings.  I am curious about a few aspects, which I would like the authors to investigate.

(1) Overall PPV is rather small (<0.5), but I wonder if certain types of true interactions are more accurately inferred by evolutionary coupling analysis?

(2) Is PPV really high for certain families or structures?  It seems to be the case, as there are many outliers in Fig. 2 (for example), and it will be great to see some discussion of this pattern.

(3)  Also, the improvement over the raw scores is very encouraging, but do the correction types show any difference among structures? That is, do all methods produce high PPV for the same types of structures?  

Author Response

Response to reviewer 2:

Comments to the Author

It is a very useful study to benchmark the performance of various methods for detecting evolutionary couplings.  I am curious about a few aspects, which I would like the authors to investigate.

We thank the reviewer for their time and comments and address each point individually below.

(1) Overall PPV is rather small (<0.5), but I wonder if certain types of true interactions are more accurately inferred by evolutionary coupling analysis?

One important point that we wish to clarify is that our goal was not to assess/maximize the overall accuracy of these methods in terms of their PPV. For instance, we down-sampled all of our alignments to 1,000 sequences because of the sheer number of models that we tested/fit throughout this manuscript. Work by our group and many others has shown that predictions are much more accurate for numbers of sequences on the order of 10,000-100,000. Thus, many of the PPV scores that we report (<0.5) are artificially low because we are not using the fully available datasets. However, we do note that there are many occasions where only ~1,000 sequences are available and therefore this choice to only include 1,000 sequences in assessing method performance is both realistic and relevant. As for certain types of true interactions, work by our group has shown that within a protein family, side-chain atom interactions are more accurately inferred than backbone atom interactions (Hockenberry & Wilke, PeerJ, 2019: https://doi.org/10.7717/peerj.7280). In this revision, we have emphasized some of these issues in the Discussion to make this more explicit:

“Another potential area for future research is to specifically investigate the cases when sequence re-weighting makes the largest/smallest impact on PPV relative to uniform weights. The overall predictive power of evolutionary couplings varies both within and between protein families. Previous work has shown that within an individual protein family, structural contacts that are mediated by side-chain interactions are most likely to be detected by co-evolutionary methods, as opposed to those mediated by atomic interactions in the peptide backbone [18]. Between protein families, the substantial variability in PPVs result from numerous factors including but not limited to the number of sequences in an alignment, the diversity of those sequences, homo-oligomerization, alignment errors from repeat proteins, and family structural variation [56]. While we found that PPVs are highly variable across protein families and that sequence re-weighting can help increase these scores a relatively small amount on average (Figure 2), the magnitude of this increase is higher for some families compared to others. Being able to associate the magnitude of the increase with properties of the sequences or tree, such as their diversity or bias, may provide interesting clues about the general ineffectiveness of sequence weighting or insight into novel strategies that could better account for these effects.”

(2) Is PPV really high for certain families or structures?  It seems to be the case, as there are many outliers in Fig. 2 (for example), and it will be great to see some discussion of this pattern.

Indeed, the reviewer is correct that these methods work overall much better for certain protein families compared to others. Other groups have more explicitly investigated variability in accuracy across different protein families and found a number of predictors of accuracy (see for instance: https://doi.org/10.1073/pnas.1702664114). Here, our goal was not to explicitly test this variability or its origins but rather to perform our analysis on a diverse set of protein families (of which we fully expected to find that some protein families would be more accurate than others). We explicitly note this variation in the revised manuscript and refer the reviewer to the paragraph that we added in response to Comment #1.

(3)  Also, the improvement over the raw scores is very encouraging, but do the correction types show any difference among structures? That is, do all methods produce high PPV for the same types of structures? 

We agree with the reviewer that this is a particularly interesting point, and possible area for future research. Anecdotally, we found that most corrections we tested produce a moderate increase in accuracy for all protein families rather than a large increase for some and no increase for others. However, there is still some variability in the “% increase” in accuracy in weighted vs unweighted models, and we suspect that some of this variability may be related to the structure of the phylogeny. At this point, however, we are not sure how to best approach this and note it as an area for future research within the discussion, in the paragraph cited in response to Comment #1 that deals specifically with these issues of variability and it’s potential sources. 

Reviewer 3 Report

The authors experiment with a new way of down-weighting redundant sequences by considering phylogenetic information. Though the results are not very positive, they are important for further studies considering the incorporation of phylogenetic information into co-evolution models.    Some concerns: For the Phylogenetic based weights, it would be nice to see how sensitive these are to the accuracy of tree reconstruction. Given these weights are depended on the tree topology and rooting, I would suggest rerunning the tree generation with a random sampling of sites with replacement (aka bootstrap). No need to rerun CMMpred, just a simple scatter (Pearson correlation) plot would be enough between two bootstrap runs.

The rescaling of weights (max vs mean) seems to be a critical part here. Some discussion as to why one might want to do this might be helpful to the readers. (Figure 1B, makes me wonder if the weights should be log or power transformed, before rescaling by mean or max?) This leads me to the next point.

The loss function used in CCMpred is:
-sum(PLL*seq_weights) +  0.2 * (length-1) * L2(2-body) + 0.01 * L2(1-body)   The regularization strength is proportional to the number of effective sequences (Neff, which is the sum of sequence weights). The more sequences you have, the smaller the L2 penalty. The 0.2 and 0.01 parameters were empirically determined, assuming a specific weighting scheme (0.8 sequence identity). If the authors are going to change the weighting scheme, these parameters may also need to change.   It would be interesting/informative to see a plot where the authors do a scan of the "LFACTOR" parameter for each of the methods. Though I suspect the mean-rescaling may already be correcting for this.   In the discussion: "...down-weighting sequences that for instance share 50% sequence identity would make little sense (and indeed, doing so produces less accurate results..." This is not clearly shown since the weights were not rescaled by mean for the different sequence identity thresholds. For example, in the case of 10%, where the weights are effectively uniform, the performance is worse than uniform, because the regularization is much stronger.

In the discussion: "...Despite being weakly correlated with one another, uniform, 80% sequence identity, HH, and GSC weights perform roughly equivalently at predicting residue–residue contacts....". This makes be a wonder if presenting the average accuracy is the best comparison. Are there some proteins that had much higher accuracy? If the authors were to spot-check the phylogeny tree for these, does it appear more star-like or bifurcating?    

Author Response

Response to reviewer 3:

Comments to the Author

The authors experiment with a new way of down-weighting redundant sequences by considering phylogenetic information. Though the results are not very positive, they are important for further studies considering the incorporation of phylogenetic information into co-evolution models.    

We thank the reviewer for their time and thoughtful comments, and address each of these issues below.

Some concerns: 

For the Phylogenetic based weights, it would be nice to see how sensitive these are to the accuracy of tree reconstruction. Given these weights are depended on the tree topology and rooting, I would suggest rerunning the tree generation with a random sampling of sites with replacement (aka bootstrap). No need to rerun CMMpred, just a simple scatter (Pearson correlation) plot would be enough between two bootstrap runs.

We fully agree that there is a lot of uncertainty/error in alignment, tree re-construction and rooting algorithms, etc. and it seems likely that this uncertainty may explicitly affect the resulting weights from phylogenetic methods that rely on all of these steps. Following the reviewer recommendation, we performed the bootstrap analysis that they suggested on each protein and include this information in the main text with a supplementary figure. Overall, we found that GSC weights in the bootstrap replicate were fairly highly correlated with the originally reported weights (mean/median correlations around 0.85) but that ACL weights were much more variable (mean/median correlations around 0.6). This sensitivity helps to explain why ACL weights performed so poorly across all of our analyses, as they are likely very noisy in their estimation.

“We additionally note that tree-based weighting methods may be subject to numerous errors during the tree construction and rooting process. We performed bootstrap resampling of multiple sequence alignments to compare the resulting weights to the originally calculated set of weights and found that they were significantly positively correlated (Supplementary Figure 1).  GSC weights, however, were much more robust (median Spearman’s ρ of 0.84) compared to ACL weights (median Spearman’s ρ of 0.61).”

The rescaling of weights (max vs mean) seems to be a critical part here. Some discussion as to why one might want to do this might be helpful to the readers. (Figure 1B, makes me wonder if the weights should be log or power transformed, before rescaling by mean or max?) This leads me to the next point.

We agree that the re-scaling point is both extremely important and also difficult to motivate/explain to general readers. We have, however, taken care to revise some of our explanation surrounding this issue and hope that our reasoning is more clear.

“Differences in absolute scales will affect the output of co-evolutionary models because the regularization strength used during model fitting is proportional to the number of effective sequences. Which is to say, a model fit to data where all weights are assigned a uniform value of 1 will be different from a model fit to the same set of sequences where all weights are assigned a uniform value of 0.1. Thus, both the relative differences in weights and their absolute scale are important considerations. For each of the three new methods we employ two re-scaling strategies... ”

The loss function used in CCMpred is:

-sum(PLL*seq_weights) +  0.2 * (length-1) * L2(2-body) + 0.01 * L2(1-body)   The regularization strength is proportional to the number of effective sequences (Neff, which is the sum of sequence weights). The more sequences you have, the smaller the L2 penalty. The 0.2 and 0.01 parameters were empirically determined, assuming a specific weighting scheme (0.8 sequence identity). If the authors are going to change the weighting scheme, these parameters may also need to change.   It would be interesting/informative to see a plot where the authors do a scan of the "LFACTOR" parameter for each of the methods. Though I suspect the mean-rescaling may already be correcting for this. In the discussion: "...down-weighting sequences that for instance share 50% sequence identity would make little sense (and indeed, doing so produces less accurate results..." This is not clearly shown since the weights were not rescaled by mean for the different sequence identity thresholds. For example, in the case of 10%, where the weights are effectively uniform, the performance is worse than uniform, because the regularization is much stronger.

This is a very important point, and we thank the reviewer for their clear and concise explanation. We considered at various times throughout the development of this manuscript how to effectively deal with the fact that the regularization strength and effective number of sequences are intimately intertwined. As the reviewer notes, this was our initial rationale for trying different re-scaling schemes for certain weighting methods. To more comprehensively address this question in light of this comment, we have re-run an extensive analysis showing that while stronger/weaker LFACTOR values can slightly improve the accuracy of certain weighting methods, overall this does not seem to be a major confounding variable and no combination results in a higher accuracy than 0.2 with the 80% sequence identity weights. We did not perform a full sweep of this parameter for every single weighting scheme but believe that our results with several are sufficient to show that this parameter should not radically affect our conclusions.

“A caveat that we previously noted is that the regularization strength of the CCMPred model is  proportional to the effective number of sequences. The typical value used for the pairwise regularization strength parameter (‘LFACTOR’) is 0.2, but this regularization strength was tuned for the previously best performing 80% sequence identity-based weights that are commonly employed.   For the GSC, ACL, and HH methods, we tested a range of parameters (from 0.05 to 1.0) to see if a different pairwise regularization strength parameter might produce superior results (Supplementary Figure 2). No combination of weighting method and parameter values, however, result in substantially improved accuracy for the best performing APC couplings. For all of the max scaled methods, smaller values of this parameter substantially improve results but only up to the level achieved by the best performing mean-scaled methods. Perhaps most notably, for entropy-corrected couplings we found that larger pairwise regularization strength parameters were helpful for the best performing methods and brought the overall mean PPVs nearly on par with that of the APC couplings (Supplementary Figure 2). The strength of regularization is thus an important consideration when evaluating different weighting schemes, but our finding that numerous methods achieve roughly the same overall accuracy remains unchanged from this analysis.”

We also re-worded the statement in our discussion to read:

“If this is the case, down-weighting sequences that for instance share 50% sequence identity might make little sense. Fully answering this question, however, would require testing a range of regularization strengths since the effective number of sequences at that level of sequence identity down-weighting is substantially less than at 80% sequence identity.”

In the discussion: "...Despite being weakly correlated with one another, uniform, 80% sequence identity, HH, and GSC weights perform roughly equivalently at predicting residue–residue contacts....". This makes be a wonder if presenting the average accuracy is the best comparison. Are there some proteins that had much higher accuracy? If the authors were to spot-check the phylogeny tree for these, does it appear more star-like or bifurcating?    

We agree that there may be limitations to only considering the average increase in PPV/accuracy. In our initial observations, however, we were unable to find any glaring differences between trees/alignments that showed higher/lower increases in accuracy when comparing phylogenetic weighted models to uniform. Virtually all protein families showed a small increase in accuracy as opposed to the scenario where some increased a lot and some did not really change. Of course, there was variability and some families increased a higher amount than others but we suspect that a larger number of alignments/trees would be required to thoroughly evaluate this hypothesis. We include a paragraph in our discussion speculating this as a possible area for future research.

“Another potential area for future research is to specifically investigate the cases when sequence re-weighting makes the largest/smallest impact on PPV relative to uniform weights. The overall predictive power of evolutionary couplings varies both within and between protein families. Previous work has shown that within an individual protein family, structural contacts that are mediated by side-chain interactions are most likely to be detected by co-evolutionary methods, as opposed to those mediated by atomic interactions in the peptide backbone [18]. Between protein families, the substantial variability in PPVs result from numerous factors including but not limited to the number of sequences in an alignment, the diversity of those sequences, homo-oligomerization, alignment errors from repeat proteins, and family structural variation [56]. While we found that PPVs are highly variable across protein families and that sequence re-weighting can help increase these scores a relatively small amount on average (Figure 2), the magnitude of this increase is higher for some families compared to others. Being able to associate the magnitude of the increase with properties of the sequences or tree, such as their diversity or bias, may provide interesting clues about the general ineffectiveness of sequence weighting or insight into novel strategies that could better account for these effects.”